# Modeling of Information Processes in Social Networks

**Sergey Yablochnikov [1],\***[ID]**, Mikhail Kuptsov [2],\***[ID] **and Maksim Mahiboroda [3]**

1. Department of Ecology, Life Safety and Power Supply, Moscow Technical University of Communications and Informatics, 123423 Moscow, Russia
2. Department of Higher Mathematics, Ryazan State Radio Engineering University named after V. F. Utkin, 390005 Ryazan, Russia
3. The Academy of Law and Management of the Federal Penal Service of Russia, 390000 Ryazan, Russia; maks-net@yandex.ru
* Correspondence: vvkfek@mail.ru (S.Y.); kuptsov_michail@mail.ru (M.K.)

**Abstract:** In order to model information dissemination in social networks, a special methodology of sampling statistical data formation has been implemented. The probability distribution laws of various characteristics of personal and group accounts in four social networks are investigated. Stochastic aspects of interrelations between these indicators were analyzed. The classification of groups of social network users is proposed, and their characteristic features and main empirical regularities of mutual transitions are marked. Regression models of forecasting changes in the number of users of the selected groups have been obtained.

**Keywords:** social network; information dissemination; modeling; statistical regularities

## 1. Introduction

Under the conditions of the formation of the information society, socio-economic relations are significantly transformed. This is primarily due to the widespread introduction of information and communication technologies (ICT) in all spheres of activity. Numerous information resources are being actively created, computer networks are being designed, hardware and software for telecommunications and communication are being developed. The so-called digitalization has significantly changed the daily life of society as a whole, and individual citizens in particular. The attitude towards ICTs has also fundamentally changed for all segments of society, whose representatives have become users of numerous social networks which unite individuals. In fact, they have created peculiar virtual interest clubs. The first social networks as specific online platforms appeared in the 1970s and 80s and the term "social network" was first proposed and justified by John Barnes back in 1954 in a publication [1]. Such innovations further allowed individuals not only to carry out virtual communication with each other, using computer social networks, but also to conduct online conferences, implement business projects, in a certain way contribute to the intensification of political activity and much more. Over time, computer-based social networks have begun to play a significant and, in some cases, exceptional role in the life of modern man.

However, the above-mentioned activities have negative consequences as well. In particular, attempts to implement global manipulation of consciousness, as well as information confrontation (information wars) are actively implemented in virtual space. Such active destructive actions are technologically close to aggressive marketing and advertising. Their goal is to manipulate the minds of representatives of certain groups of users of information and telecommunication technologies. In this case, they purposefully construct a system of certain priorities, modify the goals, and a certain part of the information is interpreted and provided in the necessary aspects to a certain subject. All this allows those who carry out numerous information attacks to impose their assessment of events on a certain target audience, to form a certain public opinion, for example, by distributing specially selected

and systematized information, usually containing unreliable or distorted information, among representatives of such communities.

In particular, today, social networks are very successfully used all over the world as a tool for efficient organization of political, humanitarian, charitable and other actions [2–4]. It is necessary to note that, in 2020, a considerable increase in the number of users is characteristic of all social networks. This may well be due to the COVID-19 pandemic and, as a consequence, the increase in the time people spend in self-isolation near computers.

The high efficiency of the above-mentioned activities in networks is determined by the huge number of their users. Very popular networks include Instagram (one of the largest social networks in the world, in the second quarter of 2020 it had more than a billion active users) [5], Twitter (a network for public messaging between users, at the beginning of 2019 the average number of users who use Twitter or related special applications daily was 134 million, and 330 million users visited this social network at least once a month [6]), Telegram (also a very popular social, which allows the exchange of messages and media files in various formats, which in 2020 has 400 million active users [7]), and VKontakte (a social network that is especially popular among Russian-speaking users, in June 2020 the number of users only in the Russian Federation reached the level of 73 million [8]).

The huge number of users of computer-based social networks and their continuous growth predetermines the interest of the business community, various political groups, state institutions and structures, and, consequently, modern researchers of information and social processes. Thus, the analysis, modeling, and predicting the behavior of information flows in social networks, the study of stochastic regularities of this process is a fairly relevant scientific task. In particular, the results of system analysis of processes in computer social networks are used to reveal the structure, nature and essence of a set of interactions between users, predicting their behavior, classification, modeling paths and intensity of information flows distribution [9,10]. At the same time, the implementation of the analysis of changes in the behavior of social network users, as a consequence of their virtual communication, is very adequate and transparent, due to the relative availability of statistical data necessary for this [9].

The purpose of this research study is to identify and study the factors influencing the dynamics of the sale of goods and services in various social networks. In addition, the local purpose of the study is to identify and analyze the set of factors, which are characteristic of social networks, but at the same time significantly affect the social behavior of people in real socio-economic relations and implementation of political activity (for example, the formation of the set of participants of those or other real, not virtual, political actions, etc.).

## 2. Literature Review

Social networks refer to multi-agent sources of information. Agents in social networks can include users, communities, groups, etc., between which the exchange of information is realized. Note that in some scientific publications [11,12], various messages are considered as agents. These messages, in fact, are the information exchanged between users of social networks. Various types of information exchanges, mutual transitions from one state to another, as well as data on the structure, dynamics and essence of the above processes, are very valuable material for carrying out system analysis by the scientific community. However, the properties of multidimensional information processes and multi-agency information sources predetermine a set of problems and peculiarities of implementing the study of the behavior of subjects and objects in social networks and very complex relationships between them [13].

In the analysis of social networks, we can distinguish [4,13,14] two main approaches to the implementation of scientific research, namely: static and dynamic. The static approach can be applied if topological aspects of the social network under study are more important than, for example, temporal aspects [4]. In particular, this approach to research implementation can be effectively applied if changes in the state of network components and

parameters of various processes are very dynamic compared to the rate of change of the network infrastructure [15]. In turn, the so-called dynamic approach to the study of social networks is characterized by a large number of different nuances. Based on the materials of the publication [16], within the framework of the dynamic approach it is possible to distinguish the so-called "mechanistic" and statistical directions of research realization. Thus, "mechanistic" research is characterized by the identification of a set of cause-and-effect relations or the determination of the direction of vectors of the main information flows, as well as attempts to explain the evolution of the structure of social networks, etc. Statistical studies, as a rule, only state the fact of certain stochastic regularities in the internal implementation of such networks information processes [17].

Regardless of the aforementioned approaches to the realization of research processes in social networks, the most widespread is the method of realization of such analysis, which is based on the application of graph theory [10,18–26]. In this case, the graph vertices, as a rule, correspond to numerous agents, and the graph edges characterize the set of interactions (implementation of information exchange) between them. For the effective use of such methods, it is necessary to carefully study the topology of social networks, to form a classification of agents and to identify the characteristics of their interactions [27].

The dynamic methods based on the application of graph theory and corresponding models are characterized by their own specific terminology, which is somewhat different from the terminology inherent to the static methods [28–33]. In particular, the concept of "dynamic graph" acquires a specific interpretation. When studying the evolution of social networks, using the above dynamic methods, we often analyze the dynamics of graph edges change [23,34–36]. The graph edges, as well as its vertices, can be periodically active, and their characteristics can depend on both personal properties of social network users and their network activity [37,38]. In this context, a very important indicator can be a peculiar level of trust of both users to each other and to information exchange channels and directly to the content [25,39]. The algorithm for determining this level of trust was proposed in [40].

Also quite common are methods for analyzing social networks based on the application of mathematical models congruent with epidemic propagation models [10,41–45]. In these models, based on systems of differential equations, modeling is carried out using such concepts as "information infection", "information immunity", "information impact", "information counteraction", etc. [41]. The above methods are focused on the processing of mainly quantitative data. Therefore, when applying them to analyze social networks, researchers have to solve serious problems associated with the formalization of numerous qualitative parameters. It should be noted that problems similar to those mentioned above are typical of the majority of currently known methods for analyzing social networks [46,47]. Since information exchange for a part of agents can occur periodically, it is quite interesting to search for possible periodic solutions and stable integral manifolds of the corresponding systems of differential equations [48]. In this case, stable integral manifolds will be interpreted depending on the simulated situation. In particular, when modeling the dissemination of information in social networks, it can be social groups, resistant to negative (positive) information influences.

In addition to the above methods, we should also mention quite original researches based on building different models of social networks: triangle-closing model [49], forest fire model [50], random forest [51], support vector machine [52], regression models [17,53], method based on convolutional neural networks [54]. Various methods and models of social network analysis and their practical application are presented in more detail in [10].

The above review of scientific literature shows that a significant part of the research related to modeling and predicting processes in computer social networks is devoted to the analysis of their internal evolution and all sorts of internal trends. However, in our opinion, the aspects of interconnection of processes (virtual) that take place in the internal space of social networks and real processes in the external environment are of the greatest interest. For instance, researchers [18,19,28,55–57] have considered a number of issues related to the

dissemination of information in such networks, taking into account the social aspects of the information transmitted.

At the same time, no less important is the study of the essence and forecasting of the effect of the state of internal space of computer social networks and virtual processes within them on real socio-economic relations and political processes in society [6,7,17,41,58–64]. A special role in these studies is assigned to the analysis and modeling of the so-called "mass" processes, in particular those that concern the collective behavior of computer social network users with consequences for the entire society and state. In particular, this applies to various political processes: from elections to protest actions [58–63]. The importance of such processes for the development of the European and world community is so great (as evidenced by the events that took place in a number of countries, from Belarus to the United States) that their study becomes not just relevant, but archival, especially in view of their permanent evolution. Of interest are a number of publications [60–63] in which researchers have analyzed a set of factors endogenous to social networks and identified mechanisms of influence on political processes implemented outside social networks. In all the aforementioned studies the authors drew attention to the fact that such influence definitely exists objectively.

The authors have established a significantly higher integration of citizens into the political processes "intra" social networks (in the so-called "online-processes") compared to the real political processes ("offline-processes") [62,63]. First of all, this is due to the presence of low-active participants of "offline processes" and certain groups of the population. Such a fact can be attributed to the presence of social network users' confidence in the effectiveness of "online activities". However, the level of activity of social network users does not correlate with the type of "online actions" (elections, public meetings, volunteer movement, crowdfunding, etc.).

It should be noted that political "online processes" and "offline processes" are essentially interrelated. At the same time, two parallel processes develop in social networks: "the dynamics of recruitment, and the dynamics of information diffusion" [61]. The velocity of information diffusion is influenced by the global reach of the audience and the topological location of information dissemination nodes within the social network. The "dynamics of recruitment" is mostly influenced by the personalization of connections within the network (so-called "reciprocal connections") [61].

The authors of the publications [61,63] put forward the hypothesis of a significant influence on the political processes, which are implemented both in online and offline formats, of the so-called "active" user groups. It is the "active" user groups that mainly determine the velocity of dissemination of information about upcoming political events and actions, as well as the rate of integration of new supporters into them. However, in our opinion, some peculiarities of the implementation of the mechanisms of interconnection of political processes in online and offline formats have not been sufficiently studied in the scientific literature. In particular, the relationship between the number of participants in political "offline processes" and the probabilistic characteristics of information dissemination in social networks that are characteristic of political "online processes" have not been analyzed. One of the aggregate goals of this article is to study the above-mentioned relationships.

A more detailed review of studies such as those mentioned above can be found in [61]. In this work, the authors have analyzed the probabilistic regularities and specific characteristics of information dissemination processes in computer social networks during time intervals from a day to a year. In doing so, they use various mathematical and statistical methods, including regression modeling. In particular, these researchers presented results that allow developing the ideas outlined in earlier publications [17,41,48,58]. Since modern political technologies based on the use of ICT tools are congruent with the technologies of aggressive marketing [65], we put forward the following hypothesis: "Endogenous factors of social networks, which largely determine the dynamics of selling goods and services in virtual space in the conditions of general digitalization of social and economic processes,

are likely to a certain extent congruent with the factors that determine the implementation and outcome of political processes".

## 3. Materials and Methods

With the help of the resources https://allsocial.ru/communities, https://app.jagajam.com/ru, https://popsters.ru, https://smartmetrics.co, https://trendhero.io/ru (accessed on 21 October 2020), statistical data were downloaded, indicating the implementation of processes in 68 communities and content of personal accounts in four social networks, namely: "VKontakte" (34 communities and accounts), "Instagram" (21 communities and accounts), "Telegram" (11 communities and accounts), "Twitter" (2 communities and accounts). Further, five communities with closed statistics from "VKontakte" were additionally monitored daily for four months for content publications. In addition, statistical data for the benefit of the authors of this article were provided by the owners of 6 accounts (3 from "VKontakte", 3 from "Instagram"). Also, the community "Youth Fever" in the social network "VKontakte" was specially created by us for the empirical impact on the population of users and for testing various statistical hypotheses and, as administrators of this community, we had full access to current statistics. The functioning of the "Youth Fever" community was maintained, in fact, for five months. Thus, in the end, in the process of implementing the study we carried out a systematic analysis of data on 75 different communities and personal accounts in four quite popular social networks.

The methodology of data uploading differed depending on the stages of the research. At the initial stage of the study, we identified four types of communities and individual accounts in social networks with the conventional names "individual", "blogger", "marketing" and "political". We classified as "individual" those personal accounts in which more than 80% of publications serve only for communication and presentation of themselves by users in virtual space. We classified as "blogger accounts" those personal accounts and communities in which more than 80% of the content serves to increase the number of "views" of the content in order to obtain further advertising contracts (the ultimate goal being to gain personal profit). By "marketing" we meant those personal accounts and communities in which more than 80% of publications serve to increase the volume of sales of certain goods and services.

Finally, we have classified as "political" those personal accounts and communities in which more than 80% of the postings have an agitation-propaganda nature and are primarily aimed at attracting the maximum number of supporters of certain ideas and attention to the activities of a political group or a single politician. Such content is quite often used for obtaining, ultimately, certain electoral benefits, in particular, during the campaigns for the elections to the state administration bodies of various levels. In our opinion, it should be noted that there are also mixed-type communities, the placement of content in which simultaneously pursues several goals, united within a single information policy or some paradigm (for example, web-pages and accounts in the social networks of the media).

Thus, most media outlets, by publishing certain content, very successfully try to realize a combination of goals. This is both selling their content and getting advertising contracts. Also, as a rule, by publishing certain content, they support one or another political group (party). In this study, we did not consider the processes in the communities and the corresponding content of personal accounts of the mixed type, because we have not found any such community with open statistics and we could not get permission from the owners of such information resources to access the statistical data of any community with closed statistics. Finally, we also did not study "individual" type communities, as this did not fit the purpose of our research.

In the first phase, the selection was based on an ordered list of VKontakte communities, available at https://allsocial.ru/communities (accessed on 21 October 2020). Then, 400 communities were randomly selected, of which 208 were immediately excluded from the list of communities to be analyzed, due to the presence of a very large number of

bots in them. Of the 192 communities remaining after the preliminary analysis from the essence, we selected those that met the criteria for classification as "bloggers", "political", "marketing" and "mixed". From the set of "blogger" communities selected in this way, we randomly selected 9 (all of which had open statistics), and among the "political" communities, 5 with open and closed statistics, respectively.

For all of the selected "marketing" and "mixed" communities it was typical to have "closed" statistics, which did not allow to analyze the data on the volume of sales of goods and services. Therefore, the administrators of these communities were sent emails inviting them to participate in our study in exchange for transparency in all of its results. Six "marketing" communities accepted our offer, and they gave us access to the relevant statistics. Unfortunately, we did not receive consent from any of the administrators of the "mixed" type communities to cooperate in the research.

Thus, general statistical patterns of information processes within social networks were studied on 26 group and personal accounts. For the 18 social network "VKontakte" accounts under study with open statistics, "unloading" data was implemented, showing the numerical values of the following indicators: total number of visits; number of unique visits; number of subscribers; number of so-called "likes", number of "reposts", number of comments, reach and reach of subscribers. For another five political communities with closed statistics on the social network "VKontakte" we "unloaded" statistical data (unloading statistics data), showing the numerical values of the majority of all of the above indicators, except for the number of unique visitors, the coverage and the scope of subscribers. Six "marketing" groups additionally "unloaded" statistical data corresponding to the numerical values of orders for goods and services. For another three "marketing" Instagram groups, we were unable to obtain statistical data regarding reach, subscriber coverage, number of unique visits and "reposts". It should also be noted that for the three communities, additional statistical data were uploaded showing the number of content publications, in particular "posts", photo and video content. The data about the number of publications for the three "marketing" groups of the social network "Instagram" were also "uploaded". In the end, the above indicators were normalized relative to the number of subscribers for the previous day.

The following remarks should be made. Each user of a computer-based social network will be referred to one of four non-overlapping sets (classes): "visitor", "subscriber", "subscriber-visitor", "adept". The "adepts" set will include those users of the social network who are consumers of goods or services of the "marketing" community, or who participate in real actions related to the agitation and propaganda activities of political communities, including the so-called protest actions. A "subscriber" is a user who is appropriately subscribed to the news of a network community, but who has not visited the community's web page on a given day and is not an "adherent" of that community. A "subscriber-visitor" is a user who has a subscription to community news, but who, unlike a "subscriber", has visited the community's web page on a given day, but who is not an "adept" of the community.

A "visitor" is a user of a social network who on a given day has visited a community web page, but who does not belong to the multitude of "adepts" or "subscribers" of that community. Note that any user of the social network who visited the community's page on a given day can be assigned to one and only one set (class). Thus, the union of these sets (classes) is the entire set of users who visited the community page on a given day, taking into account the set of "subscribers" who did not visit the community page on that day at all. The intersection of the above sets (classes) is an empty set.

The classification proposed by the authors allows us to adjust the goal of this study, defining it as the identification and systematic study of the totality of factors influencing the number of "adherents" of communities in social networks. Note that the main purpose of "political" and "marketing" communities is to increase the number of "adherents", while for "blogger" communities such a goal cannot be formulated in principle.

Separately, it should be noted that the number of "visitors", "subscribers" and "subscribers-visitors" can be obtained directly as a result of analyzing the results of statistical data download for all of the above communities. However, it makes sense to point out the fact that such "uploads" are not quite identical to the sets (classes) of social network users we suggested earlier. Thus, in order to adequately determine the number of "users" from the set (class) of "subscribers" it is necessary to subtract the "subscribers-visitors" of that community from the number of subscribers established according to the obtained statistical data. To determine the number of users from the set (class) of "subscribers-visitors", use such statistical indicator as "coverage of subscribers". To determine the number of users from the "visitors" class, subtract the number of "subscribers-visitors" from the total number of unique visits according to the statistical data.

In turn, to determine the number of "adherents" for marketing groups it is necessary to use the data regarding statistical indicators of the number of "orders" of goods and services. It should be noted that the "order" can be realized by both "subscribers-visitors" and "visitors", so the number of "orders" is subtracted in different cases from the number of different classes of users. However, the number of "adherents" for political groups could not be determined either by analyzing the data obtained by "unloading statistics data" or by analyzing the data corresponding to the daily monitoring of processes in such communities, because none of the social networks we studied have (and cannot have in principle) the relevant data.

Therefore, in order to study the dynamics of the number of "adherents" of political groups, we analyzed information processes in social networks over long periods of time, caused by three mass political actions in Belarus and Russia. The first period, from 13 July 2019 to 30 September 2019, which was characterized by protest actions related to the elections to the Moscow City Duma (Russia). Second period, from 09 August 2020 to 08 November 2020, which was characterized by political actions related to the presidential elections in Belarus. The third period, from 11 July 2020 to 18 October 2020. In this period, protests took place connected with political processes in Khabarovsk Territory (Russia). It should also be noted that the list of objectives of this study did not include further prediction of the number of participants in these actions.

In order to obtain reliable information about the number of protesters, we analyzed 127 sources of information (64 in Minsk, 35 in Khabarovsk, 24 in Moscow), including both official data (data of the Ministries of Internal Affairs in Russia and Belarus respectively), and data provided by various media outlets. Some news agencies cited their own data, based on investigations by their correspondents and various public organizations. The mathematical expectation was used as a rough estimate of the number of participants in the actions on each specific date. The numerical value of this parameter was determined on the basis of a statistical analysis of all the data available on a certain date.

Then, as a result of a preliminary analysis of the functioning of the social networks "VKontakte", "Instagram", "Telegram", and "Twitter", we identified 49 accounts, which were characterized by a relatively high number of views of publications related to the protest actions. Statistical data on these accounts was "downloaded" and subsequently used to analyze the set of factors affecting the dynamics of the number of "adherents" of political communities. To find the factors that influence the number of "adherents" of political communities, we downloaded statistical data showing the number of subscribers, visits, comments, "reposts", publications, and "likes".

To identify statistically significant factors that affect certain variables, we used the appropriate tools of SPSS and MS-Excel software, in particular, we synthesized equations of linear and nonlinear pairwise regressions, as well as multiple linear regression. The authors considered the following five types of nonlinear pairwise regressions: logarithmic, exponential, quadratic, cubic, and inverse. Seven variables were analyzed: "relative number of visits", "relative number of subscribers", "relative subscriber reach", "relative number of orders" (for marketing communities), "relative number of likes", "relative number of reposts", and "relative number of comments". The first four variables, of the above, could

be both dependent and independent ones on a separate case-by-case basis. The other three variables were defined by the authors as independent ones. If any of the variables was defined as a dependent one in a pairwise regression, the other six variables were analyzed alternately as independent ones. In the case of multiple regression analysis, the authors considered all possible combinations of independent variables. The statistical significance of the regression coefficients was checked using Student's test, and the significance of the equation as a whole was determined according to Fisher's test.

To prove the statistical significance of differences between the selected groups of communities, we created interlinking tables. When community groups were compared in pairs, $2 \times 2$ tables were used, and when all three groups were compared simultaneously, $2 \times 3$ tables were used. If community groups were compared according to the criterion of correspondence between empirical probability distributions and theoretical distributions, the rows of the tables displayed information on the number of correspondences (or discrepancies) between empirical probability distributions and theoretical distributions. If the number of significant regressions was used as a criterion for comparing groups of social network communities, the rows of the tables displayed information regarding the number of significant and non-significant regressions. The decision about the statistical significance of differences between community groups was made on the basis of Pearson's criteria. SPSS and MS-Excel software products were used in this case.

To compare empirical probability distributions with theoretical ones, the Kolmogorov–Smirnov agreement criterion was used. The test of this criterion was implemented with the help of SPSS software product. A detailed description of the pretreatment of the data for the purpose of further application of the Kolmogorov–Smirnov criterion is given further in the text of the article.

## 4. Results

First of all, a statistical analysis of the correspondence of empirical distributions of the daily relative number of visits to community pages, the relative reach of subscribers, the relative number of "likes" ("likes"), "reposts", comments, the daily relative change in the number of subscribers of 26 communities and personal accounts ("blogger", "political" and "marketing") in VK and Instagram social networks was conducted. Empirical distributions were compared with three theoretical ones: normal, uniform and exponential. The minimum volume of samples was 50 (samples were formed from statistical data on fifty or more consecutive days following one another). We considered two time periods: from 50 to 100 days (conditional name: "short time period") and over 100 days (conditional name: "long time period").

We obtained the following results. In short time periods, the variables analyzed are characterized by empirical distributions close to normal probability distributions (see Table 1) in 58.4% of cases. This relative number is significantly higher than the relative numbers of fit to uniform and exponential probability distributions (7.8% and 3.4%, respectively). On long time periods the authors failed to prove the correspondence of sample statistical distributions to theoretical probability distribution laws in more than 65% of cases (Table 2). Consequently, the analyzed indicators are most often characterized by probability distributions that are close to normal. This is especially true for the distributions of the number of relative visits, the relative number of "reposts," and the relative "reach" of subscribers analyzed over short periods of time.

The number of so-called "views" of content is one of the most common characteristics of communities in any social networks, and data on the number of such "views" can be uploaded for virtually any community (with both closed and open statistics). Therefore, first of all, we will consider in detail the number of relative visits.

The absence of a normal probability distribution of the set of parameters on long time periods can be explained by the accumulated change in the sample estimates of the mathematical expectation and variance. However, then, for short time periods, the proba-

bility distribution should also be normal. However, in practice, this does not correspond to reality (see Table 1).

**Table 1.** Relative quantities of statistical populations for short time periods with distribution laws close to the theoretical ones.

| Name of the Variable | Relative Number of Populations Distributed According to the Normal Law | | | | Relative Number of Populations Distributed According to a Uniform Law | | | | Relative Number of Populations Distributed According to the Exponential Law | | | |
|---|---|---|---|---|---|---|---|---|---|---|---|---|
| | **1** | **2** | **3** | **4** | **1** | **2** | **3** | **4** | **1** | **2** | **3** | **4** |
| Relative number of visits, % | 68.3 | 63.6 | 50 | 62.8 | 9.76 | 9.1 | 0 | 7.5 | 0 | 9.1 | 5 | 4.3 |
| Relative change in the number of subscribers, % | 31.7 | 54.3 | 40 | 41.7 | 4.9 | 5.7 | 0 | 4.2 | 0 | 0 | 0 | 0 |
| Relative coverage of subscribers, % | 85.4 | 95 | 66.7 | 87.5 | 9.8 | 0 | 0 | 6.3 | 0 | 0 | 33.3 | 1.56 |
| Relative number of "likes", % | 78.9 | 66.7 | 20 | 61.5 | 13.2 | 21.2 | 0 | 13.2 | 0 | 9.1 | 10 | 5.5 |
| Relative number of comments, % | 57.9 | 76.7 | 0 | 51.2 | 0 | 23.3 | 0 | 7.9 | 0 | 6.7 | 0 | 2.3 |
| Relative number of reposts, % | 65.8 | 72.7 | 0 | 65.3 | 2.6 | 18.2 | 0 | 9.3 | 0 | 18.2 | 0 | 8 |
| Relative number of orders (for marketing groups), % | – | – | 15.8 | 15.8 | – | – | 0 | 0 | – | – | 0 | 0 |

**Table 2.** Relative numbers of populations for long time periods with distribution laws close to the theoretical ones.

| Name of the Variable | Relative Number of Populations Distributed According to the Normal Law | | | | Relative Number of Populations Distributed According to a Uniform Law | | | | Relative Number of Populations Distributed According to the Exponential Law | | | |
|---|---|---|---|---|---|---|---|---|---|---|---|---|
| | **1** | **2** | **3** | **4** | **1** | **2** | **3** | **4** | **1** | **2** | **3** | **4** |
| Relative number of visits, % | 45.45 | 12.5 | 11.7 | 23.8 | 0 | 0 | 0 | 0 | 0 | 8.3 | 0 | 3.2 |
| Relative change in the number of subscribers, % | 4.55 | 21.7 | 0 | 9.7 | 0 | 4.4 | 0 | 1.6 | 0 | 0 | 0 | 0 |
| Relative coverage of subscribers, % | 68.18 | 93.3 | 0 | 76.3 | 0 | 0 | 0 | 0 | 0 | 0 | 0 | 0 |
| Relative number of "likes", % | 42.86 | 33.3 | 0 | 27.4 | 0 | 16.7 | 0 | 6.5 | 0 | 0 | 0 | 0 |
| Relative number of comments, % | 47.62 | 66.7 | 0 | 40.7 | 0 | 19 | 0 | 6.8 | 0 | 0 | 0 | 0 |
| Relative number of reposts, % | 28.57 | 33.3 | 0 | 30.4 | 0 | 8.3 | 0 | 4.4 | 0 | 12.5 | 0 | 6.5 |
| Relative number of orders (for marketing groups), % | – | – | 0 | 0 | – | – | 0 | 0 | – | – | 0 | 0 |

The authors hypothesized that the difference between the probability distribution and the so-called normal distribution for short time periods was due to the advertising campaigns in social network communities. To test this hypothesis, we initiated and put into practice an advertising campaign in the "Youth Fever" group. This experiment showed that, taking into account the statistical data corresponding to the days of the advertising campaign, all statistical distributions of the number of relative visits did not agree with the normal probability distribution, and without taking such data into account, they were in good agreement.

Then we randomly selected one of the communities with low level (50) of correspondence of empirical distributions of the number of relative visits to the normal probability distribution law for short periods of time. As a result of the study of these populations, we were able to identify only five so-called "outliers" (which amounted to less than 3 of all data), whose removal from the populations that were analyzed contributed to the data transformation and classification of probability distributions as normal for short periods of time. It should be noted that the delineation of data with respect to time periods, in this case, was not implemented randomly, but based on the overall structure of the data. Of course, the so-called "outliers" may not always be due to advertising campaigns. They may well be a consequence of characteristic features of empirical distributions of statistical indicators of social network communities. However, all of the "spikes" we identified were

extremely short-lived (lasting 1–2 days) and associated with a sharp increase in the number of "views" of certain content. Therefore, such an increased short-term surge of interest of social network users is, to a certain extent, anomalous and can be quite easily explained by the placement of situationally relevant information.

We believe that, at least for the variable "number of relative visits", it is possible to divide the entire data set into several subsets corresponding to sufficiently short continuous periods of time, that almost always the number of relative visits to some information resource will be distributed in accordance with the normal law of probability distribution. At the same time, we will consciously buy single uncharacteristic "outliers". Therefore, when planning the development of some content in social networks, we can proceed from the condition of compliance with the normal law of probability distribution of the number of relative visits for short periods of time (if we do not purposefully implement advertising campaigns or place the relevant information, which leads to this kind of "outliers"). In other words, based on the criterion of compliance with the normal law of probability distribution, it is quite possible to successfully predict a certain minimum level of content views.

In addition, it should be noted that we actually proved significant differences in the number of correspondences of empirical distributions to the theoretical normal probability distribution for the relative number of comments and "likes" on short intervals of time when passing from one type of communities to another ("blogger", "political" and "marketing") with numerical values of asymptotic significance in the interval from 0.0000004 to 0.00005. A pairwise comparison of the number of correspondences to the theoretical normal probability distribution allowed us to prove the existence of significant differences in the above indicators for "marketing" communities (asymptotic significance values from 0.0000001 to 0.001). "Blogger" and "political" communities differed from each other only for the variable "relative number of visits" (significance 0.047) by the number of correspondences of empirical distributions to theoretical probability distributions. It is quite possible that this peculiarity of "marketing" communities may be due to the fact that we obtained 50 of the statistical data regarding this category of communities by analyzing the processes in the social network "Instagram". At the same time, we obtained the vast majority of data on the functioning of "political" and "blogger" communities by analyzing the processes of the VK social network.

As a result of comparing the characteristics of the network's communities regarding the number of correspondence to the normal distribution for long time periods, the authors managed to prove the existence of differences in the transition from one type of communities to another ("blogger", "political" and "marketing") only for the number of comments (significance 0.0001). We identified pairwise differences, in this case, also for the variable "relative number of visits" for "blogger" communities. Interestingly, the "political" and "marketing" communities are not statistically significantly different. It is quite possible to assume that the congruence of the main goals of "political" and "marketing" communities also determines the congruence of some statistical patterns. As a result of the analysis of social network communities regarding the number of correspondences to the uniform and exponential laws of probability distribution, the authors did not reveal any statistically significant differences in any of the variables under study.

The results of the analysis of statistical relationships indicate that in long time periods there are paired and multiple regressions (from linear to logarithmic) with coefficients of determination up to 0.73 in 63.6 of cases between the variables under analysis (Figure 1, Table 3). At the same time, the authors most frequently found statistically significant regressions for the dependent variable "relative visits" (70.5). For the other dependent variables, the number of statistically significant regressions is somewhat smaller and corresponds to the numerical interval from 28.6 to 66.5 (see Table 3). It should be noted that for the three social networking communities, regressions were also synthesized between the variable "relative change in the number of subscribers" and the variable "number of posts, photo and video content posted by network users per day." These regressions were also statistically significant in 66.7 of cases.

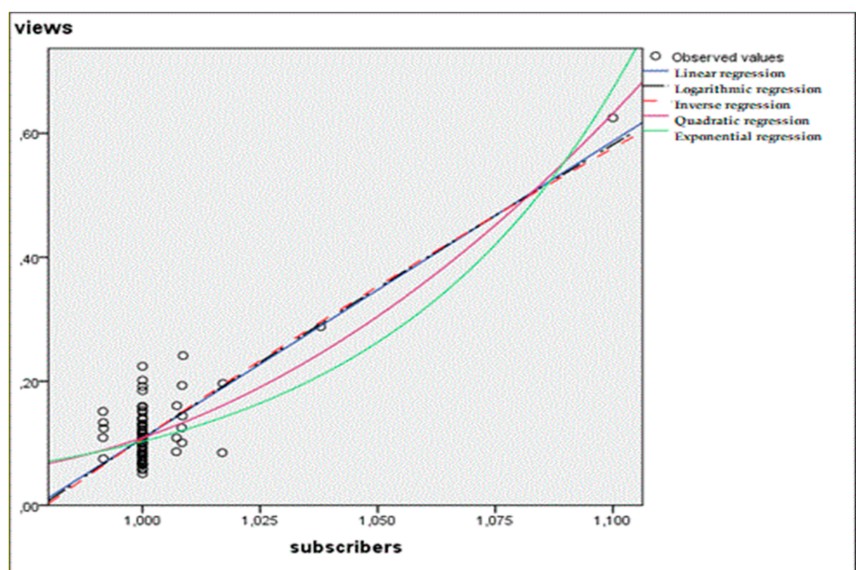

**Figure 1.** Analysis of statistical relationships.

**Table 3.** Relative number of significant pairwise linear regressions for long time periods.

| Independent Variables | Relative Number of Significant Linear Regressions for the Dependent Variable "Relative Number of Visits" | | | | Relative Number of significant Linear Regressions for the Dependent Variable "Relative Number of Subscribers" | | | | Relative Number of Significant Linear Regressions for the Dependent Variable "Relative Subscriber Coverage" | | | |
|---|---|---|---|---|---|---|---|---|---|---|---|---|
| | **1** | **2** | **3** | **4** | **1** | **2** | **3** | **4** | **1** | **2** | **3** | **4** |
| Relative number of visits, % | – | – | – | – | 100 | 100 | 82.4 | 95.2 | 68.2 | 86.7 | 52.9 | 76.3 |
| Relative change in the number of subscribers, % | 100 | 95.8 | 82.4 | 93.7 | – | – | – | – | 59.1 | 66.7 | 29.4 | 63.2 |
| Relative coverage of subscribers, % | 54.6 | 80 | 100 | 65.8 | 68.2 | 60 | 100 | 65.8 | – | – | – | – |
| Relative number of likes, % | 80.9 | 100 | 41.2 | 77.4 | 90.5 | 79.2 | 17.7 | 66.1 | 71.4 | 93.3 | 35.3 | 81.1 |
| Relative number of comments, % | 52.4 | 71.4 | 47 | 57.6 | 42.9 | 66.7 | 17.6 | 44.1 | 42.9 | 66.7 | 0 | 54.1 |
| Relative number of reposts, % | 42.9 | 66.7 | 100 | 56.5 | 33.3 | 66.7 | 100 | 52.2 | 42.9 | 80 | 0 | 59.5 |

For short time periods the regression equations were also statistically significant in 51.5 of cases. At the same time, the value of the coefficient of determination reached the level of 0.86. And the number of statistically significant regressions for the dependent variable "relative number of visits" was the greatest, 59.3 (see Table 4).

According to the authors, the above differences regarding the number of statistically significant relationships for long and short time periods are due to the fact that such interdependencies are most evident during periods of advertising campaigns implemented in social network communities. Therefore, the shorter duration of the time periods for which the values of the corresponding characteristics are estimated determines the less likely coincidence with the periods of advertising in random selection than with long ones. The same (shorter duration of time periods) can also explain the increased values of determination coefficients for short time periods. Because, given the implementation of promotional actions in some short time period, the dependence will be more pronounced than in the long one.

**Table 4.** Relative number of significant pairwise linear regressions for short time periods.

| Independent Variables | Relative Number of Significant Linear Regressions for the Dependent Variable "Relative Number of Visits" | | | | Relative Number of Significant Linear Regressions for the Dependent Variable "Relative Number of Subscribers" | | | | Relative Number of Significant Linear Regressions for the Dependent Variable "Relative Subscriber Coverage" | | | |
|---|---|---|---|---|---|---|---|---|---|---|---|---|
| | 1 | 2 | 3 | 4 | 1 | 2 | 3 | 4 | 1 | 2 | 3 | 4 |
| Relative number of visits, % | – | – | – | – | 56.1 | 64.7 | 63.2 | 60.6 | 58.5 | 80 | 66.7 | 65.6 |
| Relative change in the number of subscribers, % | 61 | 67.7 | 63.2 | 63.8 | – | – | – | – | 39 | 50 | 33.3 | 42.2 |
| Relative coverage of subscribers, % | 51.2 | 80 | 66.7 | 60.9 | 41.5 | 30 | 33.3 | 37.5 | – | – | – | – |
| Relative number of likes, % | 60.5 | 82.4 | 47.4 | 65.9 | 55.3 | 68.6 | 21.1 | 53.3 | 68.4 | 70 | 100 | 70.5 |
| Relative number of comments, % | 39.5 | 66.7 | 36.8 | 48.3 | 24.3 | 46.7 | 15.8 | 30.2 | 43.2 | 65 | 66.7 | 51.7 |
| Relative number of reposts, % | 42.1 | 79.4 | 100 | 61.3 | 31.6 | 57.1 | 33.3 | 43.4 | 34.2 | 65 | 100 | 47.5 |

In order to test this hypothesis, after carrying out an advertising campaign in the "Youth Fever" social network community, the authors synthesized regression equations based on the analysis of relevant statistical data. In doing so, two variants were considered. In the first case, the statistical data obtained for the entire analyzed time period was taken into account, and in the second case, the data corresponding to the days in which the advertising campaign was carried out were excluded from the analyzed population. Ultimately, the hypothesis formulated above was confirmed. The relative number of significant regressions synthesized using statistical data, from the totality of which those corresponding to the days of the advertising campaign were excluded, turned out to be equal to zero.

It should also be noted that the numerical values of the coefficients of determination for regressions in which the "relative number of visits" to Internet resources (for time periods of any duration) appears as a variable are usually more significant than for other variables. Note also that when synthesizing a multiple linear regression equation by inclusion or exclusion of variables, the process in 80 of cases ends up with only one independent variable, which can be quite satisfactorily interpreted. In this case, in 70 of cases this variable is "relative visits". If we consider "relative visits" as the dependent variable, in fact, in half of the cases meaningful multiple regressions have more than one (2, 3, or 4) interpretable independent variable. Thus we can conclude that the most informative is the variable "relative visits" which sufficiently combines and takes into account all other variables ("likes", "reposts", comments, various "posts", etc.).

When comparing the numbers of statistically significant regressions when moving from one type of community to another ("blogging," "political," and "marketing") we proved differences in 58.3 of cases (Tables 3 and 4). For long time periods, the corresponding level is 50, and for short time periods it is 67. For both long and short time periods, the above differences were proved for three regressions "relative number of visits from the relative number of "likes", "relative number of subscribers from the relative number of "likes" and "relative number of subscribers from the relative number of comments" (significance levels from 0.000003 to 0.038).

A pairwise comparison of the number of significant regressions for different types of communities proved differences for "marketing" types of communities in 45.8 of cases, for "political" in 28.6 and for "blogger" in 21.4. Most likely, the main factor in the emergence of the identified differences is still the fact that 50 of the data for "marketing" communities by the authors was downloaded from Instagram. However, there are also statistically significant differences between "political" and "blogger" types of communities in terms of the number of regressions in 13 of cases. Since all of the data for blogger

and political types of communities are derived from the social network "VKontakte," the differences here seem to mean that some statistical relationships are formed according to different logical schemes for different types of communities. This conclusion is also partially supported by the fact that the relative values of the deviations in the number of significant regressions for pairs of "marketing-blogger" (33.3) and "marketing-political" (58.3) types are not equivalent.

Equations of linear pairwise and multiple regressions between the relative number of orders and other variables were obtained to predict the number of "adherents" of marketing groups. For the dependent variable "relative number of visits," 47.2 of the regressions constructed were statistically significant, with 42.1 for short time periods. For the other dependent variables, the proportions of significant linear pairwise regressions ranged from 0 to 29.4 (12.1 on average). This result suggested that the number of "adherents" of marketing groups may be most closely related to the number of content views.

In order to test this hypothesis, linear multiple regression equations and corresponding equations of paired regressions of 11 kinds were synthesized with the dependent variable—"number of orders per week". The values of all independent variables were also summed up for the week (e.g., data corresponding to the variables "number of visits per week," "number of publications per week," etc. were summed up). Moreover, regression equations were synthesized both without taking into account time shifts and with such shifts for one or two time periods (one time period here is a week). This approach made it possible to generate one or two statistically significant regressions in each and every "marketing" community. In 80 of cases these are regressions without time shifts, and in 60 of cases with a shift by one time period. Only paired regressions were obtained for 80 of the marketing communities. In addition, in all cases the dependent variable was the same—"number of visits per week. And all significant multiple regressions could not be satisfactorily interpreted qualitatively (for example, they had negative coefficients with independent variables, which can be interpreted as "increasing orders with decreasing views, "likes", subscribers, etc.). Only for 20 of the "marketing" communities were formed statistically significant linear multiple regressions, which are characterized by a satisfactory qualitative interpretation. All such multiple regressions had two independent variables: "number of visits per week" and "total number of subscribers per week."

The authors' detailed analysis of the "marketing" communities allows us to conclude that the main factor influencing the increase in the number of orders is the number of content views. The influence of the above factor has been proven by the authors for all of the "marketing communities" analyzed in this study. The second most important factor influencing the results of the network communities was the variable "number of subscribers". The fact of its influence on the performance of the communities was confirmed for 20 of the "marketing" communities examined in the study. The rest of the factors we examined did not, in fact, have any effect on the change in the number of orders. However, the findings certainly require additional testing on a significantly larger set of "marketing" groups.

We will examine the results of our study of the factors influencing the number of "adherents" of political communities separately for each of the three political actions mentioned above. Let us first consider the results of the study of political actions in Moscow (Russia) in 2019. The ten communities of the social network "VKontakte" turned out to be the most active accounts promoting certain ideas and interpreting relevant events. Therefore, we downloaded statistical data only from them. We synthesized various regression equations showing the dependence of the number of participants in political actions on the values of the following variables, namely: "relative number of likes, reposts, comments, and visits. At the same time, statistical data regarding the dynamics of the values of the above variables were obtained by the authors for individual accounts. Generalized (total) indicators for the entire set of accounts or some part of them were also formed. We analyzed data for social groups three, two, and one day before and on the day of the corresponding promotion. None of the regressions synthesized by the authors were

significant. This observation applies to various types of pairwise regressions as well as linear multiple regressions.

Then a dummy variable with only two values (actually, a logical variable) was introduced, viz: «0» и«1». Equality of such a value to zero meant that the stock was not allowed by the municipal government, while one meant that it was allowed. Taking the dummy variable into account, two significant regressions were obtained: for the independent variable "amount of views the day before the action" for all ten groups with the coefficient of determination 0.80 and for the independent variable "amount of "likes" the day before the action" for all ten groups with the coefficient of determination 0.73. It should be noted that for the "political" groups, the relationship between the variables "number of views" and "number of "likes" in 85.4 of cases is shown by a significant linear regression (see Table 4).

Next, let us consider the results of the study of political actions in Khabarovsk (Russia) in 2020. Here, we encountered significant difficulties when selecting accounts for the unloading of statistical data, because very many of the data corresponding to the dates we were interested in were not available. Therefore, unfortunately, we selected for the analysis not those accounts that were more frequently "visited" by network users, but those from which the authors managed to upload data corresponding to certain dates. The data were unloaded for eight communities from Instagram, four from Telegram and four from VKontakte. As in the previous case, various regression equations were synthesized, including those containing the so-called dummy variable. Since all the actions in Khabarovsk (Russia) were not authorized by the municipal authorities, the dummy variable, taking in this case one of three values "0", "1", and "2", characterized the name of the days of the week of the action. This is due to the fact that the most mass actions were held on Saturday, fewer mass actions were held on Sunday, and the least mass actions were held on weekdays. In this case only the paired regressions containing the variables "number of "likes" in "Instagram", "number of "likes" in "VKontakte" and "total number of "likes" in "VKontakte" and "Instagram" were statistically significant. In this case, the coefficients of determination took numerical values in the range from 0.288 to 0.401. All regressions with "fictitious" variables turned out to be insignificant.

Next, we will consider the results of the analysis of statistical data regarding the implementation of information processes in social networks, caused by political actions in Minsk (Belarus) in 2020. Statistical data was downloaded by the authors on 13 communities of the social network "Instagram", seven communities of the network "Telegram", respectively, two Twitter and one "VKontakte". As in both previous cases, various pairwise and multiple regression equations were synthesized, containing all independent variables, in relation to which the statistical data was unloaded. In addition, the authors once again added a dummy variable "day of the week corresponding to the action," which could take two values: "0" and "1" ("1"—Sunday, "0"—not Sunday). Two linear regressions turned out to be statistically significant. The first is a pairwise regression showing the relationship with the independent variable "total number of views on Telegram and VKontakte" with a coefficient of determination of 0.71. The second is a multiple regression containing two independent variables describing the number of views in the two Telegram channels, with a coefficient of determination of 0.851. All regressions with "fictitious" variables turned out to be insignificant.

Summarizing the above, it can be argued that the dynamics of the number of "adherents" of "marketing" and "political" communities are primarily influenced by the number of views of the relevant content. The number of "likes" and the number of followers are also factors that influence the increase in the number of "adepts. As a result of the authors of the study, no other factors were found to influence the number of "adepts. However, we should admit that since the "number of views" is to a large extent an integral "reflection" of many other factors that we studied, it is not quite correct to ignore them at all. It is quite possible that the influence of other factors is manifested indirectly.

## 5. Discussion

Thus, the main factor influencing the increase in "adepts" of social network communities is the number of content views. This result was quite unexpected for us, because at the beginning of the study we hypothesized that the number of "adepts" should primarily correlate with the number of "subscribers" and "subscriber-visitors". We imagined the following algorithm for initiating the formation of a new "adept" in a social network community: "visitor" "subscriber" "subscriber-visitor" "adept". In this regard, we hypothesized that we would find a significantly more effective impact of the variable "number of "subscribers" of the content" on the formation of the "order portfolio" in the "marketing communities" of the network and on the dynamics of increasing the number of participants in real political actions. However, it does not seem to us entirely justified to reject the initial hypothesis. The fact is that the number of "subscribers-visitors" corresponds to the statistical indicator "coverage of subscribers", which can be downloaded only for the network "VKontakte" and only for communities of the network with open statistics. Therefore, when analyzing the factors affecting the number of "adherents" in the communities, the authors had insufficient statistical data on the variable "reach of subscribers" at their disposal. Therefore, it is necessary to implement additional research on relevant processes in social networks.

In addition, when analyzing the totality of information processes in social networks, the authors revealed the fact that, compared to 2019, in 2020 significantly more Russian-speaking "marketing" and "political" communities "promote" their content not in the social network "VKontakte", but in completely different social networks. For example, the three Instagram "marketing" groups we mentioned in this article were created only at the end of 2019. In addition, in 2019, the majority of political communities focused on promoting political action in Moscow (Russia) carried out such actions in the social network "VKontakte". In 2020, the number of these kinds of communities became significantly less. However, it should be noted that at this time we do not have enough statistical data to confirm or unequivocally refute this fact. Therefore, it may also be the subject of further research.

One more direction of further research of information processes in social networks can be revealing of differences concerning statistical regularities of different networks functioning, including those oriented at audience differing in linguistic preferences.

The identification of regression relationships between statistical indicators of communities in social networks, as well as the correspondence of most of their distributions to the normal law of probability distribution, determine the possibility of synthesizing effective simulation models of information processes in social networks and their successful application for forecasting. Such models are based on the implementation of a system analysis of possible situations and scenarios of events, in this case, in social networks.

According to the authors, the main promising direction for the implementation of further research is the synthesis of effective models for the purposeful dissemination of information in social networks and the behavior of users of such networks. At the moment, the authors have already carried out scientific research in this area, which simultaneously focuses on statistical analysis of the structure of social network communities and simulation modeling of the dynamics of user migration between their various groups, taking into account the latent (unobservable) states. As a goal of synthesis of such models we defined the identification of stable statistical dependencies for long and short time periods, which will allow to determine with a high degree of probability the trends in the development of social network communities and mechanisms of realization of such evolution.

## 6. Conclusions

As a result of the research, it was established that the absolute majority of the empirical distributions analyzed by the authors were quite close to the normal probability distributions. This is especially valid for the variable "relative number of visits" of social network users to certain web pages for short time periods. Moreover, the variable "relative

number of visits" turned out to be the most informative variable (a kind of "integral" representation of other variables). Therefore, it is quite logical to hypothesize that, in most cases, the number of "adherents" of social networking communities tightly correlates with the variable "number of visits to relevant content".

When studying the factors influencing the number of "adherents" of social network communities, we obtained an important result, which consists in the effective identification of a set of factors influencing the results of information processes implementation, as well as in the synthesis of statistically significant regression equations in all cases without exception. The above actions were implemented by the authors even in relation to statistical data due to the political actions in Khabarovsk, which are characterized by a small sample size and a certain subjectivity of the selection. Therefore, the results obtained by the authors allow us to conclude about the possibility of planning, forecasting and modeling the activities of both "marketing" and "political" communities of social networks. At the same time, despite the identified differences in statistical patterns in the transition from one type of communities to another ("blogger", "political" and "marketing"), we still found more common patterns of information dissemination in social network communities than differences. This, however, does not mean that the identified differences can be ignored in the synthesis of relevant models. On the contrary, it is necessary to take them into account and use them for more effective modeling. In particular, it seems that when modeling the processes of information dissemination in "marketing" communities, a single time period should be equal to one week, and for other communities, to one day.

Thus, the results obtained as a result of the research are important both in themselves and for further study of the statistical regularities of information dissemination in social networks.

**Author Contributions:** Resources, M.M.; Writing—original draft, M.K.; Writing—review & editing, S.Y. All authors have read and agreed to the published version of the manuscript.

**Funding:** This research received no external funding.

**Institutional Review Board Statement:** Not applicable.

**Informed Consent Statement:** Not applicable.

**Data Availability Statement:** Not applicable.

**Conflicts of Interest:** The authors declare no conflict of interest.

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
