# Peer review of "Modeling of Information Processes in Social Networks"

_information, doi:10.3390/info12030116_

Round 1

Reviewer 1 Report

Brief summary. In order to study the dynamics of information dissemination processes on social networks and their further prediction, the authors investigated the functioning of four categories of communities ("marketing," "political," "blogger," "individual") in the context of real events in Russian public life. Empirical distributions of probabilities of a set of statistical indicators were analyzed, which reflect the activity of users indicated above in the social networks VKontakte (34 communities and accounts), Instagram (21 communities and accounts), Telegram (11 communities and accounts), Twitter (2 communities and accounts), for short (up to 100 days) and long (more than 100 days) time periods. The presence of stable normal, uniform and exponential distributions of probabilities of this set of statistical indicators, as well as statistically stable connections between them, was revealed. These dependencies can be used in the future to interpret the dynamics of the evolution of information processes in social networks, as well as to synthesize regression behaviors of participants in various communities (in particular, "political" and "marketing") social networks. Factors have been identified that influence the number of participants in various real public actions and the volume of sales of goods and services in the virtual space.

1. The authors, in my opinion, ambiguously interpret the variable "number of visits per week" (as dependent or independent). In particular, the authors use this variable value in the synthesis of regression equations, in which the variable value "number of orders per week" (for "marketing" communities) appears as a dependent variable. However, the variable "number of visits per week" is hereinafter called by the authors in some cases dependent, and in others independent. In my opinion, in the text of the article, the authors need to clarify the status of the variable "number of visits per week," indicating unequivocally whether it is an independent or dependent variable.

2. The text of the article does not sufficiently define the essence of the so-called "dummy" variables that are used by the authors in regression equations. The mechanism of their influence on the number of "adepts" (for "political" communities) when analyzing events in Minsk and Khabarovsk is also not justified. To adequately understand their degree of importance, it makes sense to clarify whether the authors have experimentally established the presence of such dependence or not?

General comments. This scientific study is quite relevant. This is due to the active use of information and communication technologies and social networks to manage real processes in the socio-economic sphere, not only in Russia, but throughout the world. The value of this study in using real data to generate regression models that indicate the dynamics of social network users during the period of various actions. However, it would be very interesting to carry out similar studies using more extensive data, including those obtained on the functioning of social networks communities in different regions.

Author Response

Dear colleagues. Your comments to the article were taken into account by us and the following changes were made to the text of the article:
1. The content of the section "Introduction" was divided into two parts, namely: "Introduction" and "Literature Review".
2. In the "Literature Review" section, we added an analysis of scientific publications on the topic of this publication (paragraphs 8, 9, 10 of the "Literature Review" section)
3. In the "Materials and Methods" section, we added a description of methods for processing statistical data (paragraphs 15, 16, 17 of this section).
4.The "Results" section in the penultimate paragraph has been corrected. In particular, earlier in the text of the article, the following was stated: «The first is a pairwise regression showing the relationship with the dependent variable "total number of views on Telegram and VKontakte" with a coefficient of determination of 0.71. The second is a multiple regression containing two dependent variables describing the number of views in the two Telegram channels, with a coefficient of determination of 0.851.». This text is corrected to read as follows: "The first is a pairwise regression showing the relationship with the independent variable "total number of views on Telegram and VKontakte" with a coefficient of determination of 0.71. The second is a multiple regression containing two independent variables describing the number of views in the two Telegram channels, with a coefficient of determination of 0.851."
5. In the section "Results" (in the last three paragraphs) we stated that the dummy variables introduced in the study of political events in Khabarovsk and Minsk are insignificant.
6. The content of the section of the article " Discussion "was divided into two parts, namely:" Discussion "and"Conclusions".
7. In the section of the article "Conclusions", we added another paragraph (see the first paragraph).

Reviewer 2 Report

The paper is providing a new approach regarding the researches on social media attitudes. Modeling of information processes in social networks in Russia is a challenging job indeed.

The authors designed the research well. The abstract is fair and comprehensive.

The introduction would be accepted after the literature review part is extracted and separated in an individual chapter. In its current form, the literature review is part of the introduction chapter. I strongly suggest establishing a real analytic and critical overview of sources, in an extended form.

The material and methodology part is unusual long. However, it should be improved as the most important content, the description of the methods used, is very short, just one paragraph. We can read about sampling methods for pages, but on how the authors investigated the samples, only in one paragraph. So this part should be improved too.

There is no conclusion and no limitation chapters in the paper. The current 4. The discussion chapter should be divided into a discussion and a conclusion chapter.

After these improvements, I recommend the paper for publication.

I suggest English proofreading before re-submitting, in order to provide tha right English quality. MDPI provides such service for the authors.

Author Response

Dear Colleagues. Your comments to the article were taken into account by us and the following changes were made to the text of the article: 

1. The content of the section "Introduction" was divided into two parts, namely: "Introduction" and "Literature Review".

2. In the "Literature Review" section, we added an analysis of scientific publications on the topic of this publication (paragraphs 8, 9, 10 of the "Literature Review" section)

3. In the "Materials and Methods" section, we added a description of methods for processing statistical data (paragraphs 15, 16, 17 of this section).

4.The "Results" section in the penultimate paragraph has been corrected. In particular, earlier in the text of the article, the following was stated: «The first is a pairwise regression showing the relationship with the dependent variable "total number of views on Telegram and VKontakte" with a coefficient of determination of 0.71. The second is a multiple regression containing two dependent variables describing the number of views in the two Telegram channels, with a coefficient of determination of 0.851.». This text is corrected to read as follows: "The first is a pairwise regression showing the relationship with the independent variable "total number of views on Telegram and VKontakte" with a coefficient of determination of 0.71. The second is a multiple regression containing two independent variables describing the number of views in the two Telegram channels, with a coefficient of determination of 0.851."

5. In the section "Results" (in the last three paragraphs) we stated that the dummy variables introduced in the study of political events in Khabarovsk and Minsk are insignificant.

6. The content of the section of the article " Discussion "was divided into two parts, namely:" Discussion "and"Conclusions".

7. In the section of the article "Conclusions", we added another paragraph (see the first paragraph).

Round 2

Reviewer 1 Report

In this revised form, the article can be published.

Reviewer 2 Report

After the improvements, I can accept the paper for publication. The authors made all necessary modifications.